# Luminescent Properties of (004) Highly Oriented Cubic Zinc Blende ZnO Thin Films

**DOI:** 10.3390/ma12203314

**Published:** 2019-10-11

**Authors:** Narcizo Muñoz-Aguirre, Lilia Martínez-Pérez, Severino Muñoz-Aguirre, Luis Armando Flores-Herrera, Erasto Vergara Hernández, Orlando Zelaya-Angel

**Affiliations:** 1Instituto Politécnico Nacional, Sección de Estudios de Posgrado e Investigación, Escuela Superior de Ingeniería Mecánica y Eléctrica Unidad Azcapotzalco. Av. Granjas No. 682, Colonia Santa Catarina, Del. Azcapotzalco, CP. 02250 Ciudad de México, Mexico; lafloresh@ipn.mx; 2Departamento de Física del Centro de Investigación y de Estudios Avanzados del IPN, C.P. 07351 Ciudad de México, Mexico; lmartin@ipn.mx (L.M.-P.); ozelaya@fis.cinvestav.mx (O.Z.-A.); 3Unidad Profesional Interdisciplinaria en Ingeniería y Tecnologías Avanzadas del Instituto Politécnico Nacional, Av. IPN No. 2580, Col. Barrio La Laguna Ticomán, C.P. 07340 Ciudad de México, Mexico; 4Benemérita Universidad Autónoma de Puebla, Facultad de Ciencias Físico Matemáticas, Av. San Claudio y 18 sur, Col. San Manuel (CU), Puebla, Puebla, C.P. 72570, Mexico; smunoz@fcfm.buap.mx; 5Instituto Politécnico Nacional, UPIIH, San Agustín Tlaxiaca, 42162, Hidalgo, Mexico; erasto99@hotmail.com

**Keywords:** highly oriented crystals, ZnO thin films, spray pyrolysis technique, photoluminescence, wurtzite, zinc blende, optical properties

## Abstract

Photoluminescence properties of cubic zinc blende ZnO thin films grown on glass substrates prepared by the spray pyrolysis method are discussed. X-ray diffraction spectra show the crystalline wurtzite with preferential growth in the (002) orientation and a metastable cubic zinc blende phase highly oriented in the (004) direction. Raman measurements support the ZnO cubic modification growth of the films. Photoluminescence (PL) spectra of zinc blende films are characterized by a new PL band centerd at 2.70 eV, the blue emission, in addition there are two principal bands that are also found in hexagonal ZnO films with the peak positions at 2.83 eV and 2.35 eV. The origin of the 2.70 eV band can be attributed to transitions from Zn-interstitial to Zn-vacancies. It is also important to mention that the PL intensity of the 2.35 eV band of the zinc blende thin films is relatively higher than in the band present in hexagonal ZnO films, which means that zinc blende films have more oxygen vacancies, as was corroborated by means of the energy dispersion spectroscopy (EDS) measurements. PL spectra at 77 °K were measured and the 2.70 eV band was confirmed for the zinc blende films. Some PL bands of cubic films also appeared for the hexagonal phase, which is due, to a certain extent, to the similar ions stacking of both wurtzite and zinc blende symmetries.

## 1. Introduction

Zinc oxide (ZnO) is a highly versatile compound material because of its multiple industrial applications. As a semiconductor, ZnO thin films have many important applications, principally as sensors and optoelectronic materials [1,2,3,4,5,6]. Another interesting application is related to its use as a transparent conductor [7,8]. Therefore, researching its structural, electrical and optical properties is of wide importance. For example, Martínez Pérez et al. [7] reported the synthesis of good quality ZnO thin films by means of the spray pyrolysis technique. When chemical methods are used for depositing films, such as spray pyrolysis, it is well known that the physical properties of the resultant ZnO thin films depend on the molar concentration of the solvents and the preparation conditions of the source solution [7]. Therefore, for the same molar concentrations in the spray pyrolysis technique, if the ambient conditions of the source solution are modified then the physical properties of the deposited thin films should be different. In this research work, we show an example proving this hypothesis, and consequently a phase change from wurtzite (WZ) to zinc blende (ZB) was obtained at ambient conditions in the growth of undoped ZnO thin films on glass substrates. Ashrafi and Jagadish [9] remarked on the importance of obtaining a stable ZB phase of ZnO thin films for their potential applications. Also, investigations on the growth of the cubic phase in ZnO thin films have recently been reported. Rosales-Córdova et al. [10] reported the detection of the zinc blende phase in doped wurtzite ZnO thin films. Wang Xiao-Dan et al. [11] reported on cubic ZnO thin films deposited at a low pressure by means of the molecular beam epitaxy technique. In a previous work [12], we reported the synthesis of metastable zinc blende ZnO thin films. Their structural nanometric properties and the optical band gap were studied. In relation to their optical emission properties, in this work it is shown that the traditional photoluminescence (PL) bands [13,14,15,16,17,18,19] for hexagonal wurtzite ZnO thin films are maintained and new bands appear for cubic zinc blende ZnO thin films. The luminescent properties of hexagonal ZnO as material with a wide band gap has been widely studied in a very recent review [20], in which a large variety of electronic and optoelectronic devices were described. Cubic ZnO, with a band gap of approximately the same value and with very similar physical properties as those studied in this work, can cover the same applications. ZnO offers the prospect of replacing the gallium nitride used in most opto-luminescent devices, with the advantage of being a cheaper material. To date, ZnO nanorod-based light-emitting lasers have been reported, for which enhanced stimulated emission from very small size structures has been demonstrated [21]. Hybrid nanostructures have been fabricated by means of ZnO and graphene, which emit white light [22]. Jin et al. [23] reported violet and UV from ZnO thin films deposited on sapphire by using the laser ablation technique. Additionally, optical oxygen gas sensing by means of exciton emission detection extends the optical applications of this semiconductor [24]. Doping is also used to improve some properties with a specific purpose, as is the case where ZnO:B widens the range of luminescent emission [25], and for Zn_1-x_Cr_x_O where Cr increases the optical conductivity with respect to the undoped material [26].

The principal aim of this work is to show a new PL band of the cubic films at 2.70 ± 0.05 eV when compared with the hexagonal phase emissions, and the possible relation between the band and the structural properties of the material. The emission at 2.7 eV (λ = 460 nm) corresponds to a light blue color, which is precisely the emission of InGaN Light-Emitting Diodes [27,28]. This fact could conduce, after some engineering process, to the preparation of cubic ZnO-based blue emission devices.

## 2. Materials and Methods 

Using the procedure detailed in [7], an ultrasonic spray pyrolysis deposition system was used to synthesize ZnO thin films. Source materials solutions of 0.3032 molar concentrations of zinc acetylacetonate, from Sigma Aldrich (México), dissolved in N,N-dimethylformamide (N,N-DMF), from Mallinckrodt México, were prepared. A source solution was prepared at the same conditions (stirred for 24 h) as those reported in [7] and is denoted as solution A (Samples Z_5_, Z_6_, Z_7_, Z_8_). Another solution was prepared under different conditions (stirred for 48 h) and is denoted as solution B (Samples Z_1_, Z_2_, Z_3_, Z_4_). A spray from the solutions was produced by means of an ultrasonic generator operated at 0.8 MHz. High purity air at flow rates of approximately 10 L/min was used as the carrier of the spray to the top of the substrate for 5 minutes. To achieve pyrolysis, a molten tin bath was the substrate heater. Corning glasses were used as substrates, which were carefully cleaned using a well-known cleaning procedure [16]. The depositions were carried out at substrate temperatures of 400 °C (Z_1_, Z_5_), 450 °C (Z_2_, Z_6_), 500 °C (Z_3_, Z_7_) and 550 °C (Z_4_, Z_8_). To explore the structural properties and stoichiometry of the ZnO thin films, a field emission JEOL (Tokyo, Japan) scanning electron microscopy was used. Images of the sample surfaces and energy dispersion spectroscopy (EDS) measurements were carried out. To identify the crystalline phases, X-ray diffraction (XRD) measurements were made using a D-5000 Siemens (Germany) Diffractometer. For the photoluminescence measurements a Spectrofluorometer Rf5301 from Shimadzu (Japan) was used. Raman spectroscopy was carried out in a Horiba Jobin Yvon LabRAM micro-Raman system equipped with a He–Ne laser emitting at 632.8 nm. All the characterizations were carried out at room temperature (RT).

## 3. Results and Discussion

EDS measurements show 49.78% of oxygen (O) and 50.22% of zinc (Zn) atomic concentrations for type A samples, and 45.37% (O) and 54.63% (Zn) for type B samples, as was reported in [12]. It can be noted there was a lack of oxygen of the order of 4.63% in the crystalline network of the B samples. Figure 1 shows the X-ray diffraction patterns for the thin films A (samples Z_5_, Z_6_, Z_7_, Z_8_) and B (samples Z_1_, Z_2_, Z_3_, Z_4_). From Figure 1, the hexagonal wurtzite crystalline phase for type A films, highly oriented in the c-axe direction (002), can be observed. The B films showed high orientation in the (004) direction at *2θ* equals 44.6 degrees. Such a peak is characteristic of the cubic zinc blende crystalline phase, as reported in the literature [9,17,18,19]. The thicknesses of the hexagonal thin films are in the range of 450 ± 30 nm and the cubic thicknesses are in the 350 ± 20 nm range, as estimated by profilometer measurements.

XRD data are used to calculate the average crystallite size (d) from the full-width at half-maximum (FWHM) of highest-intensity XRD peaks applying Debye–Scherrer’s equation [29,30,31,32]:$$d=\frac{K\lambda }{\Delta \left(2\theta \right)cos\theta }$$
where *λ* = 1.5406, *Å* is the wavelength of the X-ray source, θ is the Bragg diffraction angle at a peak position, Δ(2θ) is the FWHM of the corresponding peak y and K~0.9 is a correction factor. The average crystallite sizes were estimated in the range of *d* = 14–20 nm for the hexagonal wurtzite phase (using the (002) reflection) and from 15 to 17 nm for the zinc blende phase (using the (004) reflection). The results for each sample are presented in Table 1. From the values in this table, a decrease of the crystallite size as the deposition temperature increases can be observed. No appreciable shifts of the characteristic XRD peaks were observed. The sizes of the crystallite correspond to the nanostructures formed in zinc blende films, as can be observed in the SEM image in Figure 2.

Figure 2 shows SEM images of the wurtzite thin films on the left (a), and zinc blende thin films on the right (b). The scale is 1 × 1 μm. Spheroid-like nanoparticle aggregates of 30 nm in diameter for the hexagonal wurtzite and nano-worms of 200 nm for the cubic zinc blende can be seen.

Photoluminescence (PL) measurements at RT were carried out on the ZnO thin films using an excitation wavelength of 310 nm (4.0 eV). Figure 3 shows the PL spectra of the hexagonal wurtzite ZnO thin films. It can be observed that the spectra include five bands at 2.30, 2.83, 3.34, 3.44 and 3.77 eV, respectively denoted as B1, B2, B3, B4 and B5. The principal band B2 (2.83 eV) is the most intense in accordance with the literature reports [13,14,15,16,17,18,19] on hexagonal films. The B5 (3.77 eV) band at the ultraviolet range is attributed to some emission near the band gap edge, as reported in Fang et al. [13] and Tapa et al. [33]. The B5 (3.77 eV) band was also reported by Cui et al. [34] as an unusual ultraviolet emission for ZnO crystals. The B1 (2.35 eV) band at the visible spectrum is related to the exciton hole–electron pair at a deep level (DL) of the band gap of the ZnO caused by point defects, principally oxygen vacancies [33,35,36,37,38].

Figure 4a,b shows the PL spectra for the cubic zinc blende ZnO thin films. It can be observed that the principal B2 (2.83 eV) band appears in all samples. However, a new band also appears, denoted as B6 (2.70 eV). In these figures, the absence of the B5 (3.70 eV) band for the zinc blende films can be observed. Consequently, a shift to the red of the PL spectra of the zinc blende films with respect to the wurtzite films can be observed. Detailed deconvolution of the bands can be observed in Figure 5. The B3 and B4 bands are related to the radiative recombination of donor–acceptor at the edge of the band gap, as predicted by Reynolds et al. [39]. In our previous work it was reported that the band gap of wurtzite ZnO (3.29 ± 0.03 eV) has a similar value to that of the zinc blende phase (3.18 ± 0.03 eV) [12]. Consequently, by considering the near resemblance of the ion stacking in the lattice of both structures, the origin of the B6 band could be proposed to be generated by Zn interstitials [40,41]. The energy level of Zn vacancies (V_Zn_) in hexagonal ZnO is located at 0.50 ± 0.02 eV, below the bottom of the conduction band (CB) [40], as illustrated in Figure 4c. It can be plausibly assumed the V_zn_ level in the cubic phase is located at 0.50 eV, below the respective CB—taking into account that the exact value of the center of the emission band, in both cases, depends on the measured band gap value. In this manner, the B2 band (2.83 eV) of the hexagonal phase is associated with the V_Zn_ to valence band transition [40,41] and, similarly, the B6 band of the cubic phase can be also identified with the V_zn_ with a measurement of 2.68 ± 0.02 eV at the center of the band.

Figure 5 shows the explicit deconvolution of the cubic zinc blende Z_3_ sample spectrum. The new B6 band at 2.70 eV can clearly be observed. Also, in the same figure the high intensity of the photoluminescence of the B1 and B6 bands can be observed. It is important to mention that the high intensity of the B1 band should be attributed to the lack of oxygen in the cubic zinc blende ZnO thin films, as found in the EDS measurements.

Figure 6 shows the PL intensity of the Z7 (hexagonal) and Z3 (cubic) samples carried out at 77 °K using an excitation wavelength of 325 nm. Figure 6a shows two bands at 2.42 (B7) which correspond, respectively, to green emission and to 2.90 (B8) for the hexagonal phase. Moreover, with the aid of Lorentzian deconvolution, the new B6 band at 2.70 eV for the cubic phase can again be clearly observed, as shown in Figure 6b, in addition to the B7 and B8 bands. It is important to mention that the emission of this B6 band corresponds to a blue emission.

Vibrational properties of the samples were studied by means of Raman measurements. Figure 7 displays the Z_2_ and Z_6_ spectra of the samples, which correspond to ZnO in cubic and hexagonal symmetries, respectively. The spectra look quite similar, which is in accordance with Zahn et al. [42], who established that frequencies of the cubic modification vibrational modes nearly coincide with those of the hexagonal phase and generally cannot be used for the purpose of distinguishing the two structures. In the case of the Z_2_ and Z_6_ films, certain differences can be seen in the 350–600 cm^−1^ region. First principles calculations by Wang and co-workers [43] predicted for zinc blende ZnO Raman TO and LO modes at 365 and 516 cm^−1^. Figure 7 shows (black curve) modes at 340 cm^−1^ and 525 cm^−1^ which could be associated with the TO and LO modes reported in [43]. With regard to the wurtzite phase, the theory predicts the modes A_1_^TO^, E_1_^TO^ and E_2_^high^, at 350, 371 and 401 cm^−1^, respectively. While A_1_^TO^ and E_2_^high^ were not observed in this work, E_1_^TO^ was at 414 cm^−1^, in agreement with the experimental results: 409 cm^−1^ [43] and 420 cm^−1^ [44]. This mode could be resolved by deconvolution, as shown in the inset of Figure 7a. Since the excitation beam is normal to the surface of the film, i.e., perpendicular to planes (002) and (004) of the hexagonal and cubic phases, respectively (observe inset of Figure 1), the electric field of the light is parallel to these planes. So, the excitation of the mode E_1_^TO^ (see inset of Figure 7b) is more favored in the hexagonal phase than in the cubic. Other modes observed in the hexagonal phase are B_1_^high^: 510 cm^−1^, A_1_^LO^: 501 cm^−1^ and E_1_^LO^: 508 cm^−1^ [43,45]. These three phonons could all be present to shape the band centered at 505 cm^−1^ of Figure 7, which is denoted as A_1_^LO^. The modes at around 650 and 800 cm^−1^ have been identified in hexagonal ZnO with multi-phonons [46] and surface phonons [47], respectively. In this way, our ZnO Raman results are reasonably supported for theoretical calculations and experimental reports. 

## 4. Conclusions

Cubic zinc blende ZnO thin films were grown on glass substrates by means of the spray pyrolysis technique. Raman measurements supported the growth of ZnO in the cubic phase. The structural characterization showed spheroid-like nanoparticles with 30 nm diameters for hexagonal wurtzite and nano-worm-like structures of 200 nm for cubic zinc blende films. In relation to the optical characterization, the unfolding of the principal band of the photoluminescence spectra of cubic zinc blende ZnO thin films was shown. A photoluminescence band peaked at 2.70 eV, characteristic for cubic rock-salt ZnO. This was ascribed to the exciton recombination of the hole–electron pair of the forbidden band gap that presented in the fabricated zinc blende ZnO thin films. It is important to emphasize that this band corresponds to a blue emission, which opens the applicability of these films as light-emitting devices. Additionally, EDS and PL studies suggest oxygen vacancies to be present in the cubic zinc blende ZnO thin films, which promote the green emission shown by the B7 band in the PL measurements at 77 °K. It is important to mention that some PL bands of cubic films are also observed for the hexagonal phase due to the similar ion stacking in both wurtzite and zincblende symmetries, which commonly gives place to polytypism in these ZnO thin films, as was reviewed by Ashrafi and Jagadish [9].

## Figures and Tables

**Figure 1 materials-12-03314-f001:**
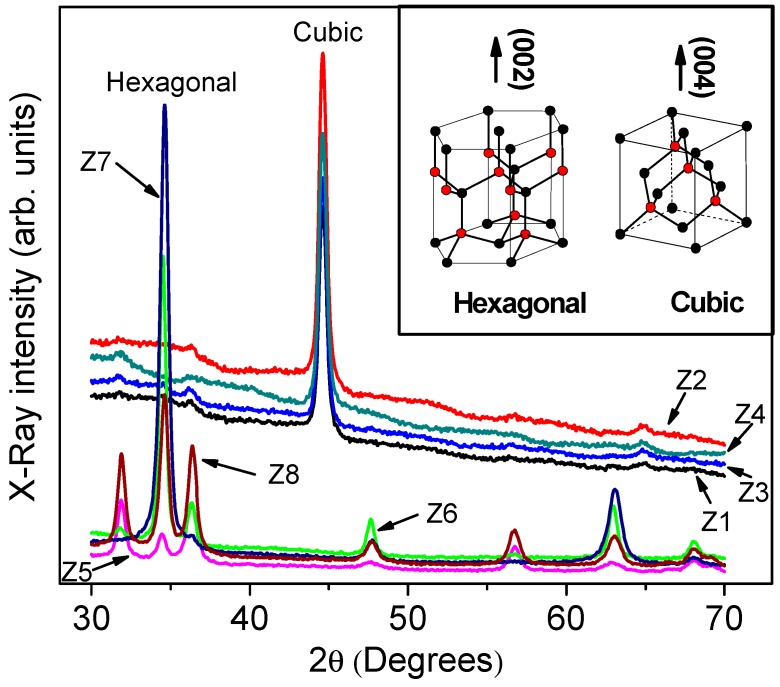
X-ray diffraction patterns of the films showing the cubic (zinc blende) and hexagonal phases. Substrate temperatures are 400 °C (Z_1_, Z_5_), 450 °C (Z_2_, Z_6_), 500 °C (Z_3_, Z_7_) and 550 °C (Z_4_, Z_8_). The inset illustrates the lattice of both ZnO phases and their preferred orientation of the films.

**Figure 2 materials-12-03314-f002:**
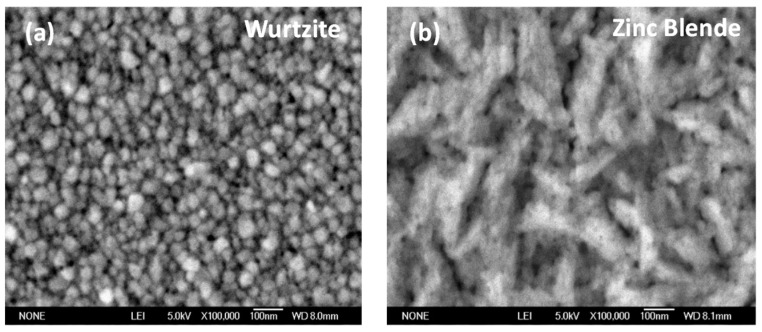
SEM images of the surfaces of the ZnO thin films. (**a**) Hexagonal phase. (**b**) Cubic phase.

**Figure 3 materials-12-03314-f003:**
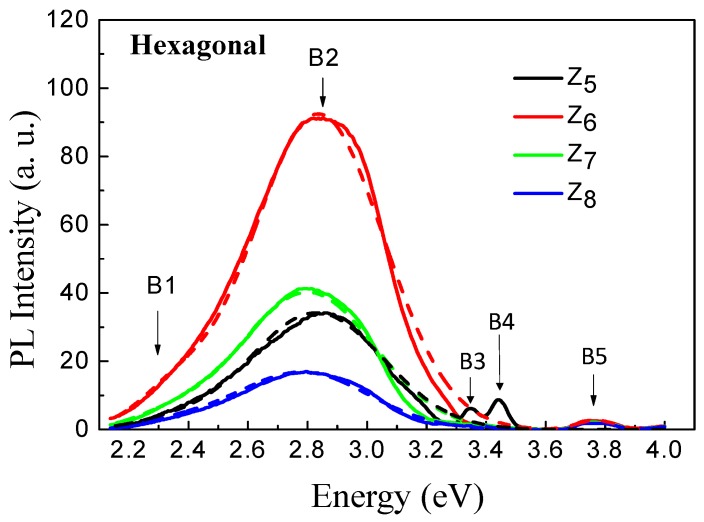
Photoluminescence spectra for the hexagonal wurtzite ZnO thin films. The dashed lines are the Gaussians, which were the references for the deconvolutions of the photoluminescence (PL) bands.

**Figure 4 materials-12-03314-f004:**
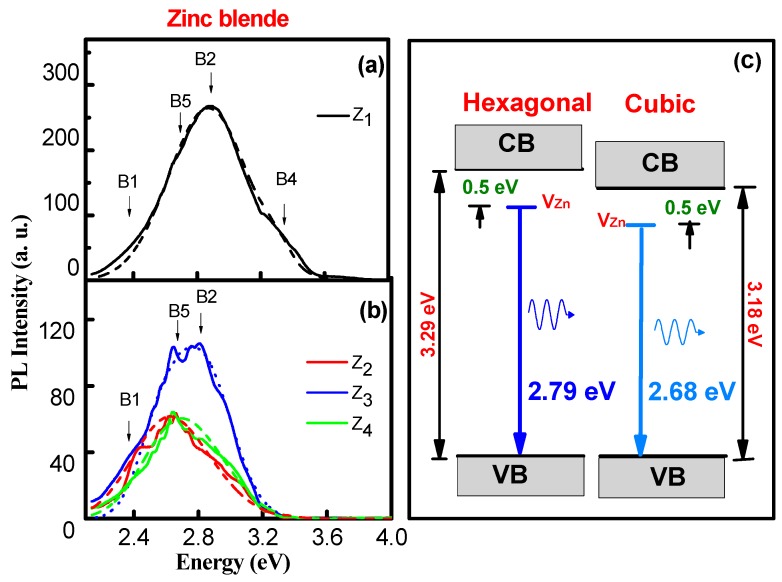
Photoluminescence spectra for the cubic zinc blende ZnO thin films. (**a**) The sample synthetized to 400 °C (Z_1_). (**b**) Samples synthetized to 450 °C (Z_2_), 500 °C (Z_3_) and 550 °C (Z_4_). The dashed lines are the Gaussians, which were the references for the deconvolutions of the PL bands. (**c**) Zn vacancy (V_Zn_) to valence-band (VB) transitions in the hexagonal (left) and cubic (right) phases.

**Figure 5 materials-12-03314-f005:**
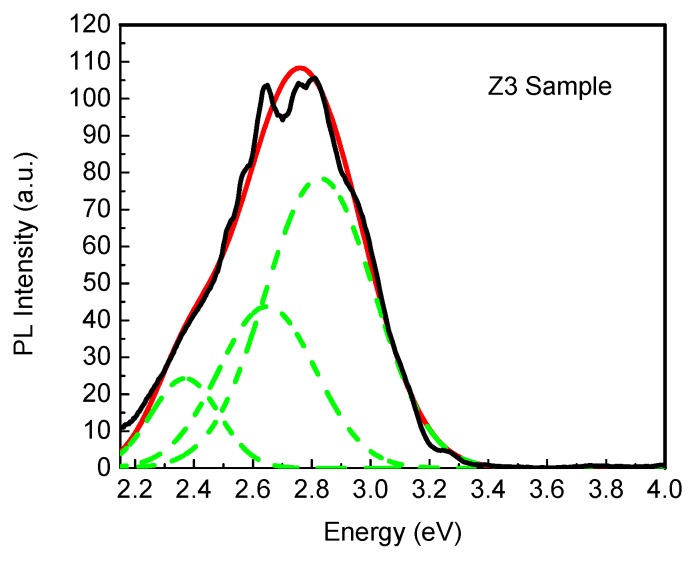
Deconvolution (green lines) of the PL spectrum of the cubic Z_3_ sample showing the B6 band.

**Figure 6 materials-12-03314-f006:**
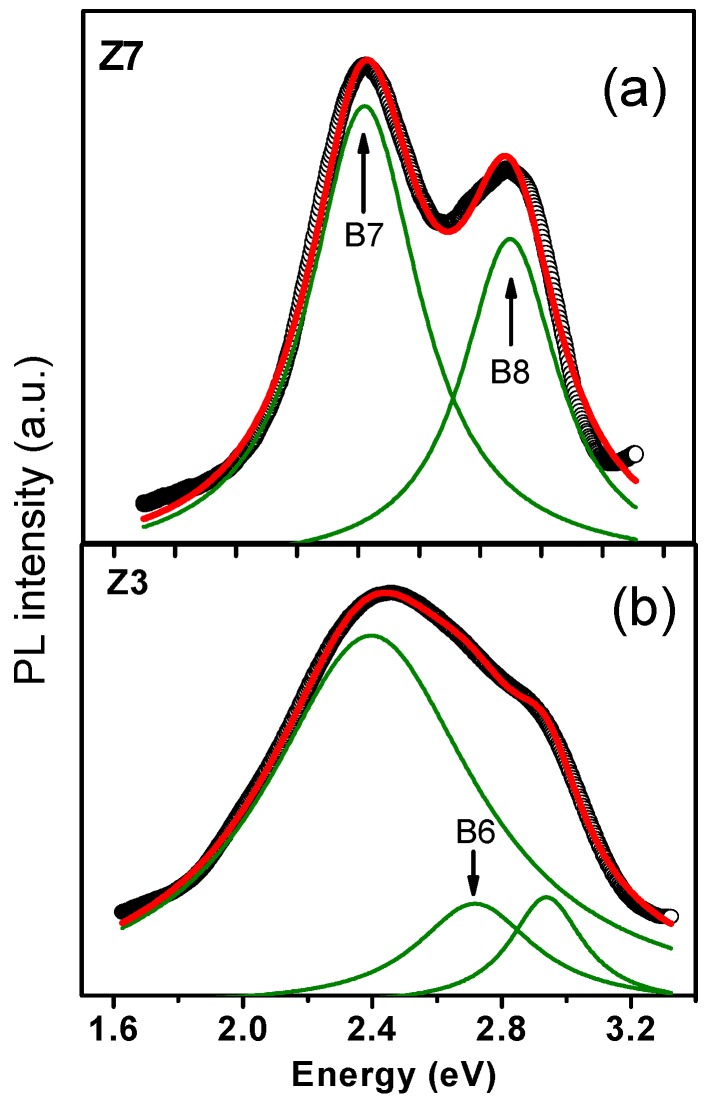
PL measurements at 77 °K showing the B6 band for the cubic phase in the deconvolution of the spectra (green lines). (**a**) hexagonal films, (**b**) cubic films.

**Figure 7 materials-12-03314-f007:**
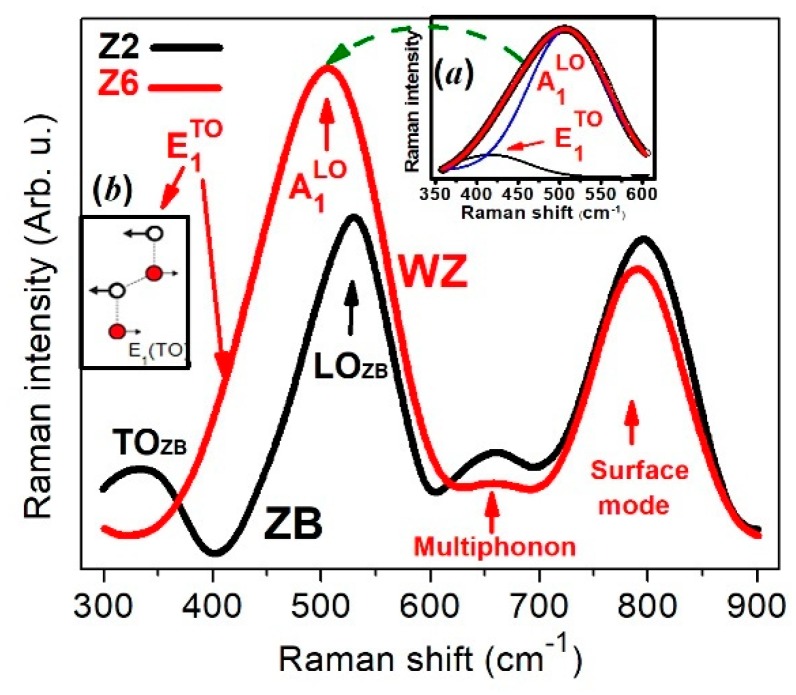
Raman spectroscopy measurements at room temperature (RT) carried out on Z_2_ (zinc blende) and Z_6_ (wurtzite) samples. Inset (**a**) exhibits a deconvolution of the band of the WZ phase in the 350–600 cm^−1^ interval. E_2_^H^ means E_2_^high^. Inset (**b**) shows the vibration of ions in the E_2_^high^ mode.

**Table 1 materials-12-03314-t001:** Crystallite size of the samples.

Phase	Sample	Full-Width at Half-Maximum (°)	2θ (°)	*d* (nm)
Hexagonal	Z_5_	0.6160	34.6200	13.362 ± 0.24
	Z_6_	0.4276	34.5014	19.452 ± 0.53
	Z_7_	0.5491	34.6088	15.152 ± 0.27
	Z_8_	0.6062	34.6200	13.762 ± 0.35
Cubic	Z_1_	0.5163	44.6148	16.629 ± 0.30
	Z_2_	0.5118	44.6186	16.777 ± 0.55
	Z_3_	0.5500	44.6200	15.613 ± 0.11
	Z_4_	0.5742	44.6000	14.953 ± 0.18

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
