# Peer review of "Luminescent Properties of (004) Highly Oriented Cubic Zinc Blende ZnO Thin Films"

_materials, 2019, doi:10.3390/ma12203314_

Round 1

Reviewer 1 Report

Authors addressed all reviewer's questions and made all requested changes. Therefore, I am glad to  recommend this paper for publishing in Materials in present form.

Author Response

Thank you for review our manuscript.

Reviewer 2 Report

The paper is well written, the conclusions are well supported by the data from the measurements. However, it is generally unclear why the exsistance of these new bands in the PL spectrum are so important and needs to be shown. It needs more extensive explanation and more justified aim.

Author Response

Thanks for review our manuscript.

Referee say "However, it is generally unclear why the exsistance of these new bands in the PL spectrum are so important " , to that respect we added the paragraph and two new references as follow: "a new PL band of the cubic films at 2.70 ± 0.05 eV as compared with the hexagonal phase emissions, and the maintained possible relation with the structural properties. The emission at 2.7 eV (l = 460 nm) corresponds to light-blue color, which is precisely the emission of the InGaN leds [20,21]. This fact could conduce, after some engineering process, to prepare cubic ZnO based blue emission devices.", we think the aim is justified.

On the other hand Referee say "and needs to be shown", to that respect a new paragraph an other two references were added as follows: "In our previous work it was reported that the band gap of wurtzite ZnO (3.29 ± 0.03eV) has a similar value to that of the zinc-blende phase (3.18 ± 0.03eV) [12], consequently, by considering the near resemblance between the ions stacking in the lattice of both structures, the origin of the B6 band could be proposed to be generated by Zn interstitials [34,35]. The energy level of Zn-vacancies (VZn) in hexagonal ZnO is located at 0.50 ± 0.02 eV below the bottom of the conduction band (CB) [34], as illustrated in Fig. 4(c). A plausible assumption can be to place the Vzn-level in the cubic phase 0.50 eV below the respective CB. By taking into account that the exact value of center of the emission band, in both cases, depends on the measured band gap value. In this manner, the B2 band (2.83 eV) of the hexagonal phase is associated to the VZn to valence band transition [34,35] and, similarly, the B6 band of the cubic phase can be also identified with the Vzn with a  2.68 ± 0.02 eV as center of the band.". Its important to mention that Figure 4 was actualized so we think it is shown the origin of the reported PL bands.

Thanks in advanced.

By the authors: Dr. Narcizo Muñoz